# An Intelligent Tutoring System to Facilitate the Learning of Programming through the Usage of Dynamic Graphic Visualizations

**Santiago Schez-Sobrino \*** , **Cristian Gmez-Portes** , **David Vallejo** , **Carlos Glez-Morcillo** and **Miguel Á. Redondo**

Department of Technologies and Information Systems, University of Castilla-La Mancha, Paseo de la Universidad 4, 13071 Ciudad Real, Spain; Cristian.Gomez@uclm.es (C.G.-P.); David.Vallejo@uclm.es (D.V.); Carlos.Gonzalez@uclm.es (C.G.-M.); Miguel.Redondo@uclm.es (M.Á.R.)

**\*** Correspondence: Santiago.Sanchez@uclm.es; Tel.: +34-926-295-300



**Featured Application: The proposal of this work focuses on offering an ITS that can be applied in programming learning environments (i.e., university degrees, schools, academies, etc.) with the objective of introducing programming transversally in different application areas and offering a complementary mechanism that facilitates its learning.**

**Abstract:** The learning of programming is a field of research with relevant studies and publications for more than 25 years. Since its inception, it has been shown that its difficulty lies in the high level of abstraction required to understand certain programming concepts. However, this level can be reduced by using tools and graphic representations that motivate students and facilitate their understanding, associating real-world elements with specific programming concepts. Thus, this paper proposes the use of an intelligent tutoring system (ITS) that helps during the learning of programming by using a notation based on a metaphor of roads and traffic signs represented by 3D graphics in an augmented reality (AR) environment. These graphic visualizations can be generated automatically from the source code of the programs thanks to the modular and scalable design of the system. Students can use them by leveraging the available feedback system, and teachers can also use them in order to explain programming concepts during the classes. This work highlights the flexibility and extensibility of the proposal through its application in different use cases that we have selected as examples to show how the system could be exploited in a multitude of real learning scenarios.

**Keywords:** intelligent tutoring system (ITS); program visualization; algorithm visualization; programming learning; augmented reality; metaphors

---

## 1. Introduction

Artificial intelligence (AI) has a brilliant present and future. In fact, there are many areas of application today with solutions to problems that society demands and that are of great interest [1]. Among them, there are sophisticated systems focused on health, driving, robotics, or even video games. A particular case in which AI has considerable importance is education, with one of its highest priority objectives the elimination of the traditional educational model currently in place [2]. This approach focuses mainly on applying cutting-edge technology to develop intelligent learning systems that simulate the interaction between the teacher and the students or that are of great value in conducting knowledge with ease.

In this context, one of the disciplines with a relevant impact is programming, as it is considered necessary to learn due to its transcendence in professional and educational contexts. In economic terms, the number of jobs that require programming knowledge is increasing [3]. The European Commission estimates that over 90% of current vacancies require programming skills and that by 2020, there are expected to be approximately 825,000 vacancies that will need to be filled in Europe [4]. From the point of view of education, programming is proposed as a transversal tool that serves as a vehicle to promote computational thinking and to work on content from other subjects [5]. Hence, the aim is to tackle problems from childhood, reinforcing concepts and developing new mental models [6]. There is, therefore, a clear need to train new professionals to cover the current demand for programmers.

However, programming poses multiple difficulties for the students who are beginning to learn it. The most important ones are the usage of variables, the comprehension of control structures, the code modularization through functions, the handling of lists, or the correction of syntactic errors, among others [7,8]. Specifically, much of this content involves a degree of abstraction in the students that they still do not have because they do not understand the real effect that it could have to change the source code of a program on its execution or because they do not provide proper solutions to problems using a programming language.

There are numerous advantages of using intelligent tutoring systems (ITS) or assistants as support tools in the context of learning how to program, since they are directed towards the development of more effective personalized teaching and learning processes [9,10]. From the perspective of the role of the teachers, the development of this type of instrument involves powerful possibilities for them, among which the reduction in time for designing content and promoting appropriate knowledge. On the other hand, from the point of view of the students, learning takes place in an autonomous way, as these systems are designed with the purpose of complementing, and not replacing, the role of the teacher.

To this approach, we can add the use of graphic representations that relate programming concepts to others with which the students may be more familiar [11]. Thus, the conversion of more complex programs into others that are more accessible and less intimidating is possible. The use of 2D visualizations has a positive effect on the learning process, from an increase in the motivation [12] and participation of the students in the class [13] to the comprehension of programming concepts [14]. Alternatively, the use of 3D graphic representations provides multiple benefits to the user, from the use of three-dimensional spaces to take advantage of spatial memory capacities, in order to understand and help to remember the structure of the programs [15], to visualizing a bigger amount of program information [16], among other possibilities. Taking advantage of the use of this type of space, we can make use of augmented reality (AR) techniques so as to visualize programs and algorithms more naturally, leading to a more fulfilling experience for the student by stimulating the activation of their brain structures or contributing to a more active learning [17].

Unfortunately, the literature lacks tutors or assistants (ITS) who present intelligent behaviors while using a representation that reduces the abstraction or complexity that a task requires. Therefore, in order to fill this gap and to improve the understanding of learning to program, we established an intelligent system to help to program by means of autonomous dynamic visualizations that guide the student during the execution of a program. The tool converts the source code of a program into a less abstract and intimidating representation, which facilitates the tracking of its trace by leaning on a computer support that generates automatic visualizations. A general scheme of the system is shown in Figure 1.

The graphic representations generated are based on a metaphor of roads and traffic signs, which have been called ANGELA (i.e., "notAtioN of road siGns to facilitatE the Learning of progrAmming"), a set of designs that establish a relationship of correspondence between programming concepts and road traffic elements. Thus, we try to define a mental model at a higher level with which to reduce the abstraction required by programming.

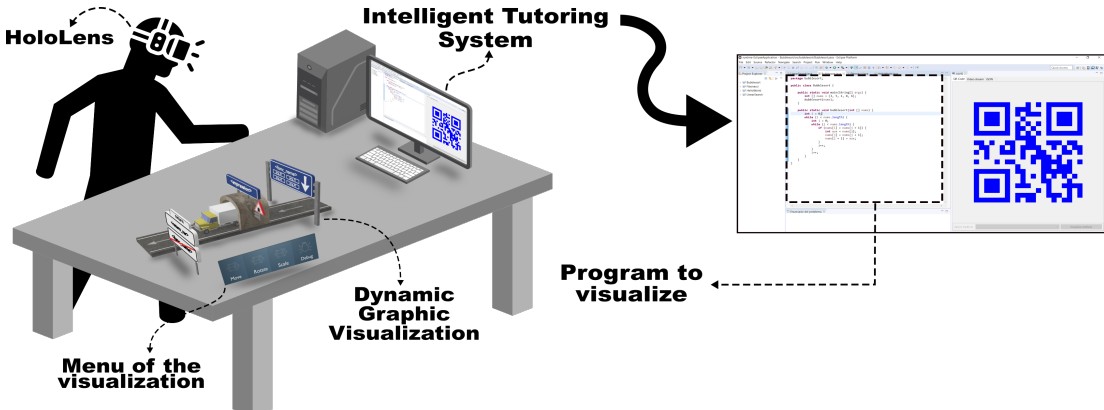

**Figure 1.** General overview of the proposed intelligent tutoring system (ITS).

The rest of the article is structured as follows: Section 2 presents some intelligent assistants (ITS) in the context of learning to program; Section 3 focuses on the proposal of this work, presents the ITS and the related integrated notation, and explains the automatic generation of dynamic visualizations; Section 4 describes a set of use cases that attempt to demonstrate the extensibility and flexibility of the proposal; finally, Section 5 details the conclusions and future lines of research.

## 2. Related Work

The existing literature presents tools to address the problem of learning to program. In this section, we review some of them focusing on both the context of intelligent assistants or tutors (ITS) and software visualization systems.

ITS is a system designed to replicate the work of teachers automatically. The goal of these systems is to provide individualized learning such as the one that a student might have by attending private classes. As described in [18], these systems are characterized by providing personalized feedback to each student, usually without requiring the intervention of a human teacher. Bearing this in mind, there are different works in the context of learning to program that are of interest.

The work presented in [19] describes an intelligent system to teach a part of the Java programming language, offering the students feedback of their solutions. Similarly, the work in [20] presented a system designed to learn how to program in Prolog language, guiding the student through an exercise. In the context of teaching web programming, the work in [21] showed a system for learning PHP, whose feature was to provide personalized feedback to the students according to their solutions, whatever they may be. Another approach was the one presented in [22], which showed a system that helped to solve custom programming exercises based on flowcharts by making decisions.

Over the years, several authors have attempted to give evidence of the benefits that ITS provides to the students. Several works conducted a literature review where they collected the results after evaluating an ITS whose conclusions confirmed the effectiveness of using this type of tool in the classroom [10,23]. However, there are studies that showed a skeptical position about the effectiveness of providing feedback to the students during the learning process. For example, in [24], it was found that the quality of feedback could lead to negative emotions such as boredom or frustration. Similarly, the work presented in [25] raised a debate about the timing of providing feedback, as this could lead to a better or worse understanding of what has been studied.

With respect to software visualization systems, there are many of them with different purposes: some focused on showing control structures and programming concepts in a user-friendly way (static visualization), and others focused on showing the trace of a running program through animations (dynamic visualization). In this last case, different systems stand out because of the graphic representations that they use.

The work in [26] described a system to represent graphically the concurrent behavior of programs using a metaphor based on cities. Along these lines, the work in [26] showed an application that used

techniques based on gamification to present puzzles in which some elements of the programming languages were treated in the form of growing plants, such as loops and iterations. Likewise, the work in [27] described a system that used avatars and 3D animations to represent programming concepts, for example exchanging messages between objects, among others. Furthermore, although more abstract, the work in [28] presented a system that helped to understand concepts related to concurrent programming through the use of characters and 3D boxes.

The literature also evaluated the effectiveness of using program visualizations and algorithms in the context of learning to program. A variety of papers presented results of tests and experiments in favor and against. Among the publications supporting this, it is noteworthy to mention the findings in [29,30], where a number of tests were held with the students who were able to demonstrate a better comprehension of the elements of the programming languages and the proposed algorithms. In contrast, the results presented in [31,32] offered a more controversial view, highlighting the use of visualizations, rather than its quality, and especially the fact that it is difficult for the teacher to create them and to keep the students engaged during classes. Besides the selection of suitable metaphors, the tools that support both the visualization and the interaction are just as important. In [33], several experiments were performed with students to discover the difficulties they had in trying to understand the problems related to recursive programming and concluded that when they used tools that helped them to visualize the behavior of the program, their ability to complete the exercises improved as it became more dynamic. Furthermore, in [34], there was a review of other related tools that use visualizations to help with the teaching of programming with successful results.

Furthermore, the effectivity of the visualizations can be enhanced by using AR techniques, as described in [35], which concluded with certain benefits that could help the process of learning to program, including improvements in learning, an increase in motivation, and easiness in interacting with one's own visualizations and collaborating with other learners; for example, the one presented in [36], in which paper based markers were used to help to teach the concepts of computer graphics programming using AR and the OpenGL 3D graphic specification; and the one presented in [37], in which the same system was assessed with the students who achieved favorable results in encouraging them to learn and enhance their motivation.

The differentiating feature of this work is the integration of an intelligent behavior into a software visualization system, which takes special care to provide feedback to the student. This work also presents another singularity: the use of AR as a visualization technology, which can be leveraged to generate learning environments on a real environment, while improving the realism of the analogy with the physical elements that it represents. Thus, we try, on the one hand, to enable the autonomous learning of the students and, on the other hand, to capture and maintain their attention and motivation.

## 3. Proposed ITS and Notation of Road and Traffic Signs

The system proposed in this paper is intended to facilitate the introduction of programming to university students who are just beginning to program and who lack sufficient knowledge to solve problems through some programming language. Mainly, this situation is linked to the abstraction required to understand the programming concepts that are taught, to the lack of knowledge of the language, or to the difficulty to design and build algorithms [7]. Thus, the proposed ITS integrates the use of metaphors that allow us to reduce this level of abstraction by including analogies between concepts from the real world and aspects of programming, trying to reduce the complexity linked to the task of programming. This analogy establishes a notation with roads and traffic signs that we have called ANGELA, which represents the structure and sentences that compose a program (static visualization), going through the representation by means of a vehicle that goes through its structure, and executing these sentences (dynamic visualization). This metaphor is intended to be motivating and easy to understand, as it includes familiar elements from the daily lives of the students. Thus, a direct analogy is established between the graphic representations in the metaphor and the sentences defined in the Java programming language, whose choice is due not only to the popularity in the professional

field, but also in the educational one [38]. The statements defined allowed us to represent graphically the function definition, the condition statement, the loop statement, the expression evaluation, and the function return. In addition, a representation was added to this set for the breakpoints and another one to simulate the execution thread of a program in order to include dynamic visualization mechanisms in the system. It is interesting to note that this set did not distinguish between different types of loops (e.g., while, for, or do-while), as well as with types of conditions (e.g., if-then-else or switch-case), simplifying the construction of the visualizations to a reduced set of graphic elements that were easily understood by the students. Figure 2 shows the set of elements that are part of the notation:

- Function definition (Figure 2a): It provides information related to the name of the function and its input arguments. This is the first graphic representation with which a full visualization must begin. The arrow of the traffic sign includes the reading direction of the visualization.
- Conditional statement (Figure 2b): It provides information related to the condition. This condition will then be used to set the new execution flow that will be followed by the program; if the condition is met, the program will execute its left side, otherwise its right side.
- Loop statement (Figure 2c): It provides information about the condition to be met to enter the loop. If the condition is fulfilled, the execution flow will be driven to the right road of the graphic representation, running all the sentences on such side. Otherwise, the execution flow will be driven through the left road. In both cases, the arrows on the road show the direction to be followed. Using a roundabout, we make an analogy with the concept of repetition.
- Function return (Figure 2d): It provides information related to the function that is finished including its name and variables to be returned.
- Evaluation of expressions (Figure 2e): It provides information related to arbitrary expressions such as assignments, function calls, etc. We use a tunnel to illustrate the evaluation of the expression when the vehicle goes through it.
- Breakpoint (Figure 2f): It provides a mechanism to stop the program execution as if it is a debugger. In this case, and with the aim of being integrated with the rest of the metaphor, we use a traffic cone.
- Thread (Figure 2g): It provides information about the current execution point of the program, allowing the user to explore the program through a functionality similar to that of a debugger. The graphic representation used, a vehicle, would go through the graphic representations step-by-step executing the related sentences. Multiple vehicles going through the visualization would represent different threads executing the same function.

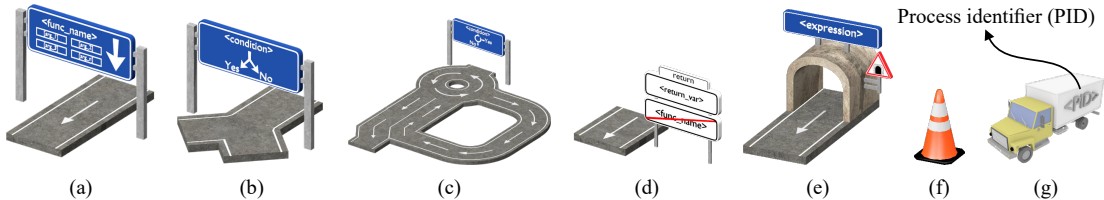

**Figure 2.** Graphic representations of notAtioN of road siGns to facilitatE the Learning of progrAmming (ANGELA) converted to an orthogonally projected 3D space. (**a**) Function definition. (**b**) Conditional statement. (**c**) Loop statement. (**d**) Function return. (**e**) Evaluation of expressions. (**f**) Breakpoint. (**g**) Thread.

The decisions about the design in the graphic representations were inspired by the "Vienna Convention on Road Signs and Signal" (1968) [39] whose proposals were adopted internationally by most countries. This presented a direct advantage to those students who were obtaining a driving license, since they were already familiar with such designs.

These visualizations could be displayed through an AR device. Among the state-of-the-art devices, Microsoft HoloLens excelled thanks to the immersion and interaction capabilities that it

offered. This way, students could benefit from using the ITS through a natural interface more intuitive for them.

The system facilitated the construction of the visualizations by generating them automatically from the source code of the programs. The visualizations generated were then sent to the AR device, which acted as a viewer that received the necessary information about the visualization from the ITS through the network. For this, from the source code of the program, a simplified hierarchy was generated in the JavaScript Object Notation (JSON (data interchange format using text to transmit data consisting of attribute-value pair [40])) format that could be easily processed by the AR device and that contained all the information necessary to construct the visualization. To do this, the ITS analyzed the abstract syntax tree (AST) generated from the source code and identified each of the language elements related to the proposed graphic representations (see Figure 2). The hierarchy generated in JSON format kept each of these associations in a tree-like form, along with the connectors needed to construct the visualization in a linked way. Figure 3 shows an example of the simplified sub-hierarchies generated for some of the proposed graphic representations. The final visualization would comprise the complete hierarchy from these sub-hierarchies.

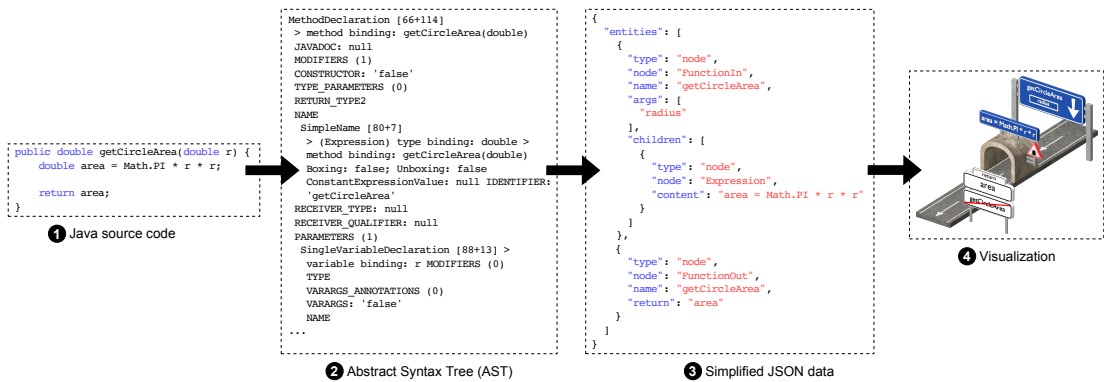

**Figure 3.** Generation of the JSON structure with the relevant information used to build the visualization from the source code of the program.

For the dynamic visualization mechanisms, the ITS was responsible for intercepting calls to the chosen debugger and sent this information to the visualization device in order to animate the movement of the vehicle through the graphic representations. On the side of the device, the interaction made by the user was sent to the ITS to manipulate the debugger status (i.e., advance the program execution, pause it, etc.).

Currently, the ITS implementation is being conducted through a plug-in implemented for COLLECE-2.0 [41,42], a collaborative distributed system for learning to program based on the Eclipse development environment. This plug-in utilizes the COLLECE-2.0 infrastructure, performing the tasks mentioned above, to generate the JSON from the AST of the source code of the program. The debugger is used by the ITS through the API provided by the JDT Debug plug-in (https://projects.eclipse.org/projects/eclipse.jdt.debug) from Eclipse.

Figure 4 shows a complete example of the dynamic visualization generated using the proposed ITS through the AR device.

The movement of the vehicle along the visualization was done by constructing a directed cyclic graph. This graph was generated from the graphic representations added to the display, which defined the set of nodes that the vehicle must follow. Figure 5 shows an example of the generation of such a graph for the visualization generated in Figure 4.

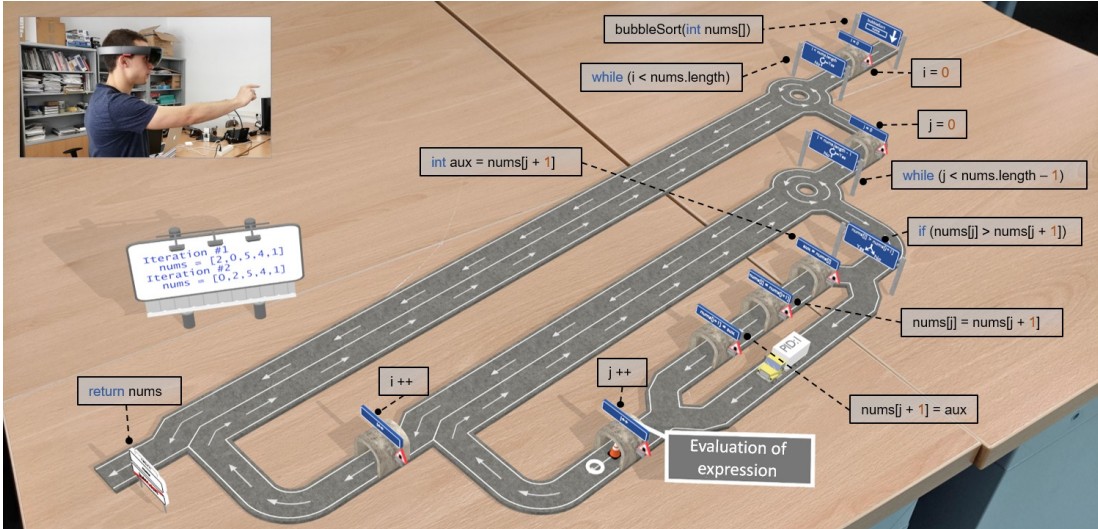

**Figure 4.** Dynamic visualization of the "Bubble sort" algorithm as seen through the AR device. At the bottom right, the execution thread of the program (vehicle) is shown moving towards the breakpoint (traffic cone) and an information panel when the cursor is superimposed on a graphic representation. At the same time, on the left side, a billboard is shown, which prints the status of the algorithm at a particular point of the execution.

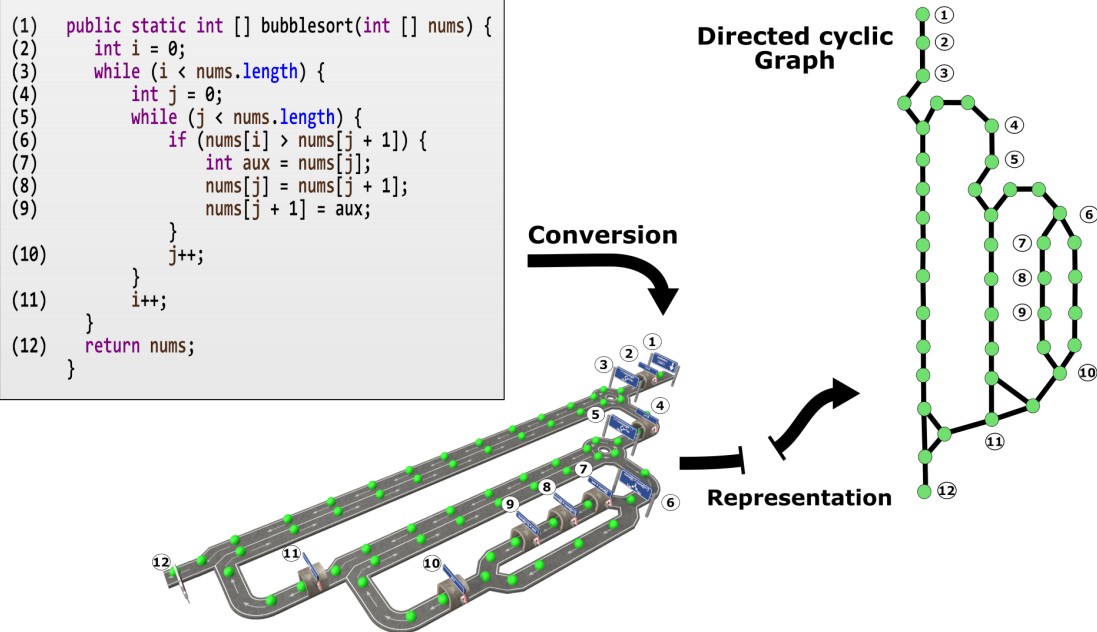

**Figure 5.** Example of the generation of a directed cyclic graph for the "Bubble sort" algorithm. The green spheres represent the sequence of nodes that the vehicle must follow.

The feedback provided to the user was used to guide and help them to understand the program that they were running. Thus, it was possible to define personalized feedback as comments in the source code, which were stored in JSON and shown to the user in the form of text bubbles that would appear when the vehicle passed through the graphic representation associated with the line of code where the comment was inserted. When defining these comments, certain control labels could be introduced to manipulate the execution of the program, as seen in Listing 1. In that case, the execution of the dynamic display would be stopped when the first comment was reached, due to the control label defined in the last line of the comment. This label system was flexible enough to define other

properties that allowed the teacher to direct the student's learning by using the comments included in the programming language itself.

**Listing 1.** Example of JAVA source code that includes a control label for the ITS.

```
1   // First, we add all the elements in the array using a reduce
2   // expression available in the Stream API introduced in Java 8:
3   //! its:execution = "pause"
4   List<Integer> integers = Arrays.asList(1, 2, 3, 4, 5);
5   Integer sum = integers.stream().reduce(0, (a, b) -> a + b);
6
7   // Second, ...
```

In addition to feedback based on comments, the ITS also incorporated a mechanism based on software metrics to provide additional information to the students. The selected metric was called "cyclomatic complexity", proposed by McCabe [43], and used to measure quantitatively the complexity of a program. To do this, we could calculate the cyclomatic complexity (CC) of a program with a single entry point and a single exit point in a simplified way according to [44], as $CC = NB + 1$, where $NB$ is the number of branches in a program.

In this context, the system calculated this cyclomatic complexity automatically so that the method was represented, counting the total number of occurrences of the following keywords: **if**, **for**, **while**, **case**, **return**, **&&**, **||** and **?** (for the conditional ternary operator). Once this value was calculated, the result was shown on the display by changing the color to red for those graphic representations that increased the cyclomatic complexity of the method over the recommended value: 10. This value was selected from that proposed by McCabe in his work, identified as a recommended limit and interpreted as the program being "simple and easy to understand".

Figure 6 shows an example of a fragment of a visualization colored in red as the cyclomatic complexity increased over the recommended value, assuming that it belonged to a larger method. In this way, the students could reconsider their solution and modify it to reduce its complexity.

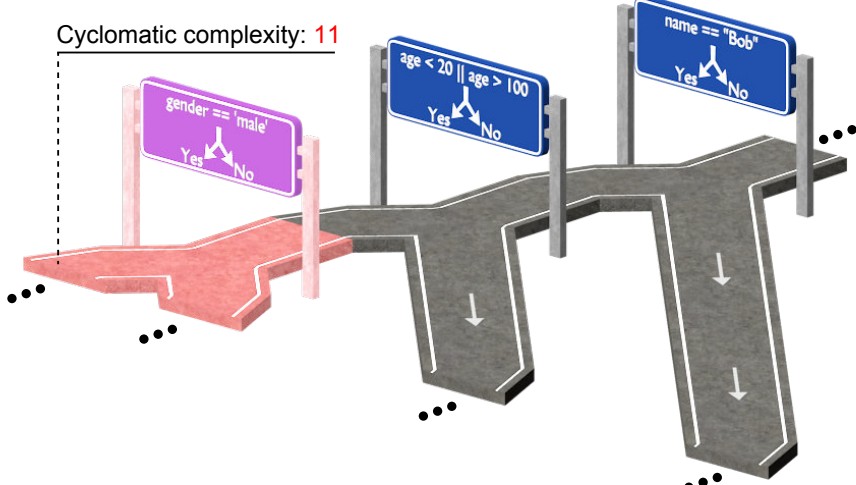

**Figure 6.** Cyclomatic complexity shown as a red shading, which highlights the graphic representation related to the statement that increases the complexity over the recommended limit.

## 4. Resulting Use Cases

The objective of the proposed ITS was to be used in learning to program environments to facilitate learning by providing aids and guides that allowed the student to understand different concepts such as program structure, control statements, their dynamic aspect by tracing the program execution, etc.

Moreover, the system could be extended to facilitate the learning of more specific programming concepts, such as those arising in concurrent programming contexts, and even in less advanced ones aimed at children who are trying to learn to program through video games. Thus, this section highlights a set of scenarios where the proposed system was used, showing its flexibility through the dynamic visualization of programs, child-oriented game environments, the use of the system as a use case in the classroom, and the understanding of concurrent programming concepts.

### 4.1. Dynamic Program Visualization

The students who were learning to program hardly used the debugging tools offered by current programming environments because their use still required a high level of abstraction, among other reasons. These tools helped to understand the execution of a program or to detect and correct errors at runtime.

In this context, the proposed ITS was used to build an interactive AR debugging environment where the teachers could teach programming concepts or demonstrate the use of a debugger in a real situation. Such an environment was built on a physical surface of the real world, from the 3D graphic representations presented in Figure 2.

Through a menu associated with the visualization, the user was in charge of starting the program debugging. Through this process, the Eclipse debugger was launched, which executed the actions performed in the visualization by sending orders through network sockets to the plug-in that acted as a server. Once the debugger was launched, the vehicle appeared, placing itself over the starting point of the visualization, which was initially the function definition statement.

The actions that the user could perform were the classic ones that a debugger integrates, which were executed through gesture interaction mechanisms supported by the visualization device. Among the most relevant actions, it is worth mentioning the possibility of checking the value of the variables when the user selected a sentence through which the vehicle passed, executing the program step-by-step through discrete vehicle movements or even adding breakpoints that enabled its movement without pausing until the statement that contained such an element. On the other hand, the user also had the possibility of checking the output of his/her program through a console in the shape of a billboard, which was placed next to the representation that printed the states through which it passed. Additionally, the comment system provided explanations to the user about what happened during the execution of the program, in order to facilitate the understanding of the source code.

This process ended when the vehicle reached the return sentence, disappearing from the display and stopping the process in both the device and the debugger in the Eclipse environment.

### 4.2. Agent-Based Game Environments

The application of the proposed ITS was interesting for programming agent-based models in game environments, in order to motivate 8 to 15-year-olds to take interest in computers and learn basic programming concepts. Students of these ages could be familiar with the concept by playing with toys reflecting actual objects, including cars, carpets simulating the streets of a city, and giant puzzles that allowed them to build structures based on roads.

The proposed system could be extended to support a didactic environment of programming based on agents, where the player communicates with the agent through a pre-established set of commands that allows them to manipulate the state of the agent.

In terms of learning, the levels of the game represented different challenges that the player must solve. These challenges were posed with a difficulty that introduced programming concepts to the player incrementally. Thus, the first levels of the game would introduce programming concepts related to the execution of expressions and conditions, to finally move on to the use of loops and functions.

The stages of the game presented a challenge to overcome by means of different actions that told the main character how to reach the goal of the level. Such stages were automatically generated over the real world by means of applying AR techniques. The character would execute the actions

sequentially, so that the player specified them as a list of commands. Thus, the player could replicate the movement of a horse in chess by specifying two orders to advance, one to turn right and another to advance forward. If the character reached the goal of the level after executing the sequence of actions, then the stage would be completed. Otherwise, the player would have to provide an alternative sequence. To check that the current action was being executed, a small character ran through them at the same time as the main character.

Figure 7 shows a snapshot of one of the levels included in the game.

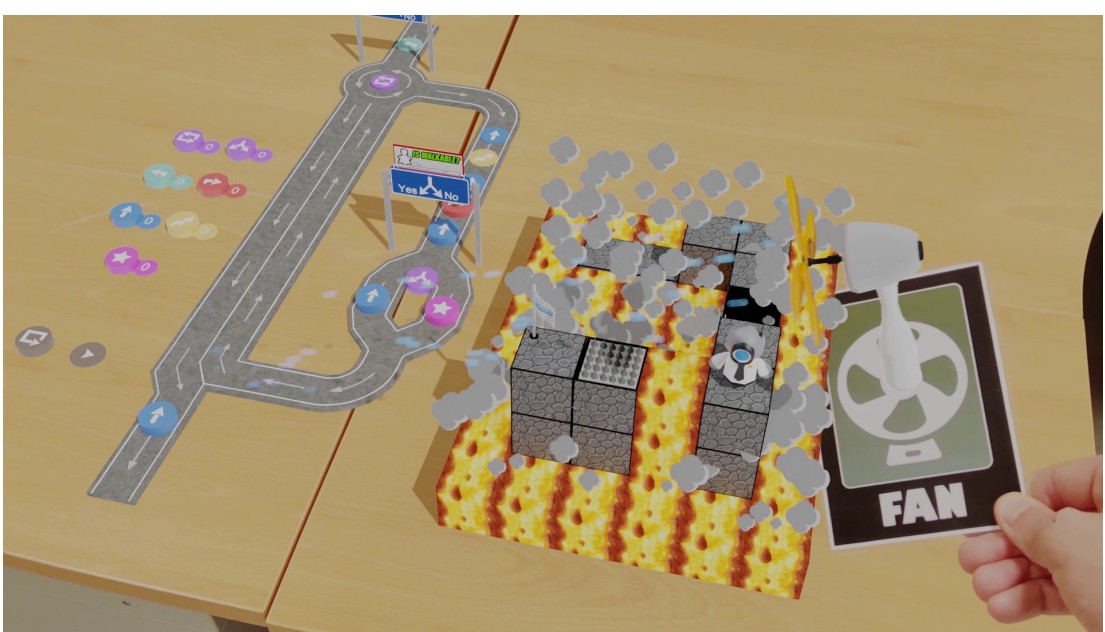

**Figure 7.** A snapshot of the RoboTIC game running the level used during the evaluation. A video playing this full level can be found at https://youtu.be/T1I2mrX7BUw.

### 4.3. Using the ITS in the Classroom

The use of ITS in the classroom was related to the use and integration in the COLLECE-2.0 environment to allow the teachers and the students to build programs collaboratively and to visualize them dynamically. This collaborative feature was achieved through COLLECE-2.0 sessions, whose actions were replicated on all visualization devices through network sockets. For this scenario, we considered that all the people involved in the learning process had this device to visualize and interact with the representation located on a real-world surface.

Mainly, the teacher was the one who decided the program to be displayed, positioning the representation on a suitable place in the classroom. In this way, the teacher shared the visualization so that the students could both interact with the representation and move freely through it.

Thus, in this use case, the ITS played a secondary role, and it was the teacher who guided the students during the visualization, stopping its execution to ask questions of the students about what was happening in the program. The students could also interact with the visualization in order to solve doubts that may arise during the lesson.

As an example and to place the description, we considered the "Bubble sort" algorithm as shown in Figure 4. In this visualization, a billboard that printed the program output was placed, a break point on an expression evaluation, and a vehicle that went through the graphic representations executing sentences.

In this context, the teacher could start an explanation by running the algorithm step-by-step during its first iterations, emphasizing the most relevant points of the execution.

Thus, the students could understand the behavior of the algorithm, using a lower level of abstraction and handling representations that were already known. Afterwards, several questions

could be asked to check the degree of understanding obtained after the initial iterations; for example, what the state of the list at a given time was, what the value of a variable in any given iteration was, or which branch of the inner conditional sentence the vehicle would pass through and why. Likewise, these questions could also be asked directly using the elements of the graphic representation.

Similarly, the students could also pose questions to the teacher about aspects that were not understood. In this case, the student could stop the execution and ask why a variable was updated to a certain value or add a ghost vehicle that served to reproduce the behavior that the main truck performed.

### 4.4. Comprehension of Concurrent Programming Concepts

A more advanced use case was the one oriented toward facilitating the learning of concurrent programming concepts. In this sense, the ANGELA notation integrated in the ITS was ideal for visualizing the execution flow of the programs, specifically through the vehicle that went through the graphic representations as a thread of execution.

Concurrent and real-time programming entailed some challenges in students who were used to working in programs with a single execution flow. This was due to the new concepts that had to be introduced to the student, such as processes, synchronization and communication mechanisms, etc. The proposed ITS could leverage the included notation to help visualize these concepts.

The identification of the critical sections in a program, i.e., memory variables that are shared between different processes, would be noteworthy among the concepts that the notation could represent. Such a concept could be represented by painting yellow diagonal stripes on the roads and placing vehicles that would represent the different running threads with their PID on their sides. In the case of the proposal, trucks were used to represent such threads.

In order to synchronize all the trucks (i.e., threads), we could use traffic lights to stop/start the movement of the trucks, that is to stop/start the running thread respectively to avoid them entering the critical section at the same time. To control the state of traffic lights, pressure plates were placed on the road to simulate the action of the wait/signal on the traffic light when the truck drove on it. Such behavior can be seen in Figure 8, where a thread with $PID = 3$ is running through the critical section. At the same time, another thread with $PID = 4$ is waiting at the traffic light before the critical section until the other thread drives on the pressure plate to signal the traffic light.

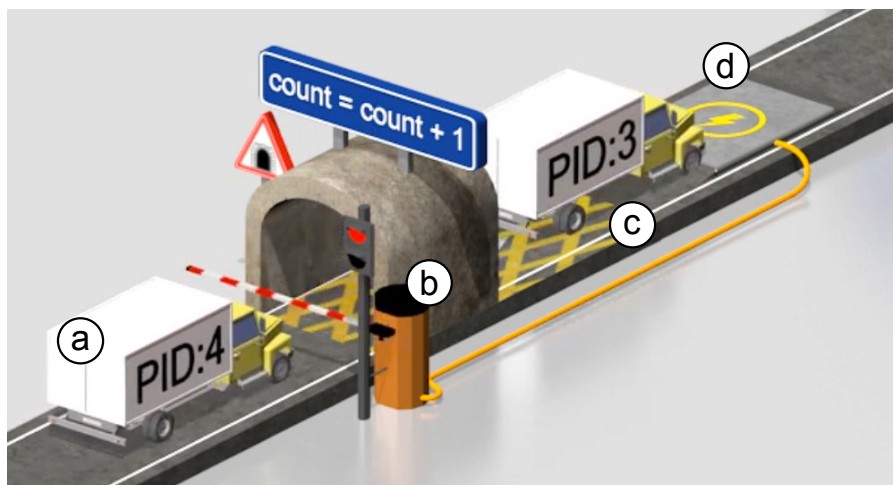

**Figure 8.** The ITS in action using the ANGELA notation showing a use case of concurrent programming. In this context, a thread (a) tries to access the critical section of a program (c), currently accessed by another thread and protected by a traffic light with a barrier (b) controlled by a pressure plate for signaling the semaphore (d). A video playing the full sequence can be found at https://youtu.be/66zaRSIjbPA.

## 5. Conclusions

In this work, we presented an ITS aimed to make learning to program easier through a notation based on the roads and traffic signs metaphor (ANGELA) through augmented reality. The system was presented in terms of the features and implementation in order to show the different characteristics of the proposal. Furthermore, through COLLECE-2.0, the system was integrated in a complete collaborative learning environment for programming.

The use cases raised exposed the usefulness, scalability, and flexibility of the system in multiple use cases, which contributed to validating the system in different areas such as interactive debuggers, its collaborative use in the classroom, the learning of concurrent programming concepts, and its integration in game environments oriented toward the learning of programming foundations in children.

As future lines of work, we highlight the need to work on two fundamental axes: (1) the in-depth study of the benefits that use cases bring to the learning of programming and (2) the analysis of their exploitation in AR environments on mobile devices with current technologies (e.g., ARCore, ARKit, Vuforia, etc.), which would allow the exploitation of the 3D space in an economic way in terms of the costs of the devices, thus giving the population with reduced resources access to the system.

**Author Contributions:** Conceptualization, S.S.-S. and C.G.-P.; funding acquisition, D.V., C.G.-M., and M.Á.R.; investigation, S.S.-S., C.G.-P., and D.V.; methodology, D.V. and C.G.-M.; project administration, M.Á.R.; software, S.S.-S. and C.G.-P.; supervision, D.V.; validation, C.G.-M. and M.Á.R.; visualization, C.G.-P.; writing—original draft, S.S.-S. and C.G.-P.; writing—review and editing, D.V. and C.G.-M. All authors have read and agreed to the published version of the manuscript.

**Funding:** This research was funded by the Ministry of Economy, Industry and Competitiveness, and the European Grant Number "TIN2015-66731-C2-2-R".

**Conflicts of Interest:** The authors declare no conflict of interest.

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
