# Peer review of "An Intelligent Tutoring System to Facilitate the Learning of Programming through the Usage of Dynamic Graphic Visualizations"

_applsci, doi:10.3390/app10041518_

Round 1

Reviewer 1 Report

A really interesting paper which would appear to have great potential for continuing research.

Two  minor points  - the acronym JSON is used but not initally explained.

The referencing technique of refereing to the number of the paper refernced rather than authors in the body of the text is unusual - is this a required convention

Reviewer 2 Report

This paper proposes to use VR to integrate in an ITS for Java programming.

The idea is very interesting. The paper is well-written and easy to understand.

I would recommend the authors two points:

1) update the references. Some are very obselet, e..g, [18, 20, 33].

2) the authors should conduct some experiments and report empirical results in a journal.

since the developed idea is not imperically validated yet, it's contribution is not sufficient to be published in a journal.

Reviewer 3 Report

This paper concerns Programming learning and  proposes
6 the use of an intelligent tutoring system (ITS) that helps during the learning of programming by using a notation based on a metaphor of roads and traffic signs represented by 3-D graphics in an augmented reality  environment. These graphic visualizations can be automatically generated from the source code of the programs, thanks to the modular and scalable design of the system, used by students by leveraging the available feedback system or used by teachers to explain programming concepts during classes. Moreover the Authors describe different use cases selected as examples to show how the system could be exploited in a multitude of real learning scenarios.

The paper is well written and very interesting. For this reason I will recommend for the publication 
